

# LeakGuard
## System IoT detekcji i zapobiegania wyciekom z domowej instalacji wodnej

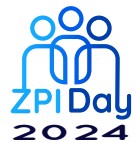

**Autorzy**: Philip Hart[1] · Szymon Jędrzejczak[2] · Łukasz Świszcz[3] · Szymon Wieczorek[4]

**Opiekun:** Piotr Jóźwiak[5]

### Streszczenie

Projekt *LeakGuard*, opracowany w ramach zespołowego przedsięwzięcia inżynierskiego w roku akademickim 2024/25, jest wielodziedzinowym projektem stanowiącym rozwiązanie problemu występowania wycieków w instalacjach wodnych. Celem opracowanego prototypu jest ograniczenie występowania wycieków w domowych instalacjach wody użytkowej oraz monitorowanie jej zużycia. W ramach projektu opracowano urządzenia IoT, oprogramowanie wbudowane, aplikację mobilną oraz małą instalację wodną służącą do testowania prototypu. Sprzęt został zaprojektowany z użyciem mikrokontrolerów z rodziny STM32 i opiera się na autorskich płytkach drukowanych. Potrzeba opracowania systemu oraz jego wdrożenia na rynek została potwierdzona podczas nieformalnych demonstracji projektu, które zainicjowały liczne dyskusje, dostarczając wnikliwego spojrzenia na pożądane cechy systemu.

## 1 WPROWADZENIE DO PROJEKTU

LeakGuard jest systemem IoT (*Internet of Things*), który służy do wykrywania i zapobiegania wyciekom występującym w instalacjach wodnych. Opiera się na dwóch modułach: urządzeniu centralnym i monitorach wycieku. Moduł centralny jest odpowiedzialny za badanie przepływu w instalacji oraz odcinanie wody w przypadku stwierdzenia wycieku. Stanowi on element instalacji wody użytkowej i jest zasilany z instalacji sieciowej budynku. Moduł monitora, który jest zasilany bateryjnie, pozwala wykrywać miejscowo zalanie i jest odpowiedzialny za powiadomienie modułu centralnego o wystąpieniu wycieku. Projekt obejmuje również aplikację mobilną, która służy do konfiguracji systemu i zdalnego obserwowania stanu instalacji wodnej. Inną funkcją aplikacji jest powiadamianie użytkownika o wystąpieniu wycieku.

### 1.1 Cele projektu

Głównym celem projektu jest opracowanie i zbudowanie prototypów dwóch urządzeń elektronicznych składających się na system IoT – modułu centralnego podłączonego do przepływomierza z elektrozaworem, a także zasilanego bateryjnie monitora wykrywania wycieków. W ramach prac należało od podstaw zaprojektować elektronikę oraz oprogramowanie obu modułów.

Urządzenie centralne ma pozwalać na pełną konfigurację systemu LeakGuard w sieci WiFi poprzez aplikację mobilną. Kryterium postawionym dla komunikacji między modułem monitora a urządzeniem centralnym była prostota, niezawodność, energooszczędność oraz możliwość pracy w warunkach komunikacji z wieloma przegrodami w budynku.

Istotnym dla projektu jest zminimalizowanie zużycia energii przez monitor wycieku w trybie uśpienia. W ramach definiowania wymagań niefunkcjonalnych ustalono, że musi on działać co najmniej 1 rok na pojedynczym zestawie baterii AAA.

Po wdrożeniu na rynek projekt przyczyni się do zwiększenia bezpieczeństwa instalacji wodnych poprzez skuteczne zapobieganie niekontrolowanym wyciekom wody. Dzięki temu ograniczy zarówno marnotrawstwo zasobów wodnych, jak i potencjalne straty materialne wynikające z awarii. W badaniach przeprowadzonych przez M. Knapika wykazano, że roczne koszty wody z niezidentyfikowanych wycieków mogą przekroczyć nawet 1700 zł [2].

---

[1]Philip Hart: Embedded Developer / Hardware Engineer

[2]Szymon Jędrzejczak: Software Engineer / QA / R&D

[3]Łukasz Świszcz: Embedded Developer / Hardware Engineer

[4]Szymon Wieczorek: Mobile App Developer

[5]Politechika Wrocławska; piotr.jozwiak@pwr.edu.pl

## 1.2  Zastosowania projektu

System LeakGuard znajdzie zastosowanie w wielu dziedzinach, gdzie istotne jest zapobieganie wyciekom wody oraz ochrona przed zalaniami. Projekt został opracowany początkowo z myślą o instalacjach wodnych w budynkach mieszkalnych, gdzie system może chronić przed zalaniem spowodowanym np. awarią pralki, zmywarki lub pęknięciem rury w instalacji wodnej. Niemniej jednak, system jest odpowiedni także dla obiektów użyteczności publicznej, takich jak szkoły, szpitale lub biurowce, w których wyciek wody może spowodować znaczące straty materialne i generować wysokie koszty napraw. Ponadto, poza funkcją zapobiegania wyciekom, system może działać jak inteligentny licznik wody, co może stanowić kolejny element ekosystemu inteligentnego domu.

## 1.3  Użyte technologie

Ze względu na interdyscyplinarność projektu wykorzystano liczne technologie sprzętowe oraz software'owe.

**Sprzęt**  Wykorzystano mikrokontrolery z rodziny STM32. Moduł centralny został wyposażony w mikrokontroler z rodziny ESP32 w celu umożliwienia komunikacji z siecią WiFi. Moduł centralny posiada 64 KB pamięci EEPROM oraz 4 MB pamięci flash. Komunikacja między monitorami a jednostką centralną zrealizowana jest w oparciu o układ LoRa SX1278 [3].

**Zaprojektowanie sprzętu**  Do opracowania schematu elektronicznego oraz płytki drukowanej wykorzystano narzędzie KiCad. Narzędzie pozwala na podstawie zaprojektowanego schematu płytki wygenerować pliki Gerber, które są potrzebne podczas procesu wytwarzania płytek. Oprogramowanie KiCad wybrano ze względu na szerokie wsparcie, zaawansowane funkcje oraz łatwość użytku.

**Oprogramowanie sprzętu**  Do oprogramowania sprzętu użyto języków C oraz C++ ze względu na ich dojrzałość i szerokie wsparcie w rozwoju oprogramowania wbudowanego. Narzędzie STM32CubeMX posłużyło do wygenerowania kodu inicjalizującego mikrokontrolery STM32. W celu zachowania czystości kodu i zapewnienia standardów kodowania, w całym projekcie zastosowano `clang-tidy` oraz `clang-format`. Kod głównego modułu opiera się na systemie FreeRTOS. Do integracji oprogramowania zastosowano narzędzie CMake, które jest powszechnie wykorzystywane w zaawansowanych projektach C/C++.

**Testowanie systemu**  Do testowania kodu oraz sprzętu zaimplementowano skrypty powłoki Bash oraz skrypty w języku Python. Skrypty te testują stabilność komponentów. Pozwoliły one przetestować wydajność kodu oraz rozwijać niektóre części projektu bez konieczności posiadania rzeczywistego sprzętu. Tę metodę wybrano ze względu na łatwość przenośnego testowania systemu i skalowalność. Elementy logiki są testowane jednostkowo za pomocą frameworka GoogleTest.

**Aplikacja mobilna**  Wykorzystano framework Flutter oraz język programowania Dart. Prototypy interfejsu aplikacji zostały zaprojektowane za pomocą narzędzia Figma, które umożliwia opracowanie projektów za pomocą gotowych komponentów. Dane konfiguracji systemu są przechowywane w bazie danych SQLite.

**Chmura**  Funkcje odpowiedzialne za powiadamianie użytkownika o wystąpieniu wycieku zostały zrealizowane za pomocą usług AWS oraz Firebase. Wybrano te platformy ze względu na prostotę i elastyczność oferowanych usług odpowiednich dla potrzeb projektu oraz możliwość prostego skonfigurowania powiadomień push za pomocą Firebase Cloud Messaging. Komunikacja urządzeń centralnych z chmurą wykorzystuje protokół MQTT. Infrastruktura chmury jest prowizjonowana dzięki narzędziu Terraform, a oprogramowanie jest konteneryzowane za pomocą Dockera.

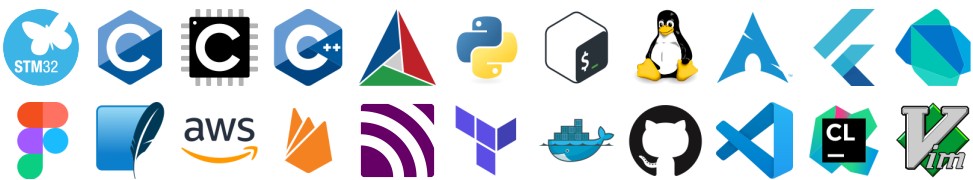

## 2  PORÓWNANIE Z ISTNIEJĄCYMI ROZWIĄZANIAMI

Większość konkurencyjnych systemów skupia się wyłącznie na detekcji wycieków lub kontroli przepływu, podczas gdy LeakGuard łączy oba te aspekty w jednym spójnym rozwiązaniu. Dzięki modułowej konstrukcji i możliwości grupowania jednostek centralnych, system oferuje większą skalowalność niż inne dostępne na rynku opcje.

LeakGuard wyróżnia się również podejściem do prywatności, przechowując dane lokalnie zamiast przesyłać je na zewnętrzne serwery, co jest standardem w innych systemach. Gwarantuje to pełną kontrolę nad danymi użytkownika, działanie bez połączenia z Internetem oraz zgodność z regulacjami dotyczącymi ochrony danych, co odpowiada na potrzeby użytkowników w zakresie prywatności.

| | LeakGuard | Flo by Moen[6] | Grohe Sense[7] | Fibaro[8] |
|---|:---:|:---:|:---:|:---:|
| Automatyczna detekcja wycieków | ✓ | ✓ | ✓ | ✓ |
| Automatyczne odcięcie wody | ✓ | ✓ | ✓ | ✗ |
| Monitoring przepływu wody | ✓ | ✓ | ✓ | ✗ |
| Bezprzewodowe monitory wycieku | ✓ | ✗ | ✗ | ✓ |
| Grupowanie jednostek | ✓ | ✗ | ✗ | ✗ |
| Harmonogramy blokowania przepływu | ✓ | ✗ | ✗ | ✗ |
| Heurystyczna detekcja | ✓ | ✗ | ✗ | ✗ |
| Lokalne przetwarzanie danych | ✓ | ✗ | ✗ | ✓ |
| Analiza zużycia wody | ✓ | ✓ | ✓ | ✗ |
| Tryb głębokiego uśpienia monitorów | ✓ | ✗ | ✗ | ✗ |
| Aplikacja mobilna | ✓ | ✓ | ✓ | ✓ |
| Praca offline | ✓ | ✗ | ✗ | ✓ |

Tabela 2.1: Porównanie projektu LeakGuard z systemami detekcji wycieków dostępnymi na rynku

# 3   NAPOTKANE PROBLEMY I OGRANICZENIA

LeakGuard to interdyscyplinarny projekt o szerokim zakresie. Problemy mogły dotyczyć zarówno sprzętu, oprogramowania, jak i zasięgu radiowego. Mimo to natknięto się jedynie na dwie główne przeszkody.

## 3.1   Serwer HTTP dla systemów wbudowanych

Podczas pracy nad projektem okazało się, że na rynku brakuje serwerów HTTP dla systemów wbudowanych, które spełniają wymagania projektowe. Rozwiązanie powinno ograniczać się do statycznej alokacji pamięci (w celu wyeliminowania problemu fragmentacji), posiadać API w języku C++ umożliwiające zaprogramowanie w prosty sposób REST API, pozwolić na wykorzystanie dowolnej implementacji stosu TCP/IP, a także zapewnić dostateczną wydajność. Najbliższym wymaganiom okazał się projekt *Mongoose*[9], jednakże zaprojektowany jest on raczej do tworzenia pełnych webowych interfejsów użytkownika, a także dostarcza API w języku C.

W projekcie zdecydowano napisać od podstaw i przetestować własną implementację serwera, której nadano nazwę *microhttp*. Implementacja ta wspiera większość standardu HTTP 1.1, dostarczając jedynie podzbiór funkcji protokołu niezbędny do realizacji projektu. Celem tego zabiegu była optymalizacja wydajności i rozmiaru oprogramowania.

## 3.2   Błędy w dokumentacji przepływomierza

W nocie katalogowej przepływomierza [8] można znaleźć informację, że posiada on wyjście typu otwarty kolektor. Podczas testów okazało się to nieprawdą – przepływomierz posiada wyjście typu push-pull. Wymagało to umieszczenia dodatkowych komponentów pomiędzy nim a modułem centralnym.

# 4   WYNIKI PRAC

W ramach projektu udało się opracować moduły i zaimplementować funkcje spełniające wymagania i cele projektu. Zaprojektowano schematy elektroniczne oraz płytki drukowane modułów. Samodzielnie

---

[6]Źródło logotypu *Flo by Moen*: https://www.ozankarakoc.com/flo-by-moen-branding
[7]Źródło logotypu *Grohe*: https://1000logos.net/grohe-logo/
[8]Źródło logotypu *Fibaro*: https://www.fibaro.com/pl/products/flood-sensor/
[9]Mongoose – https://mongoose.ws/

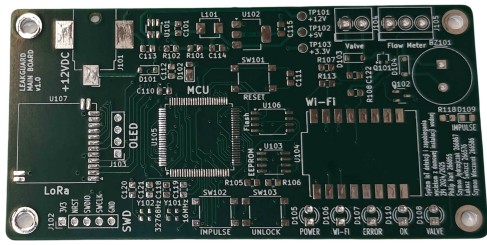

Rysunek 4.1: Gotowa płytka drukowana modułu centralnego.

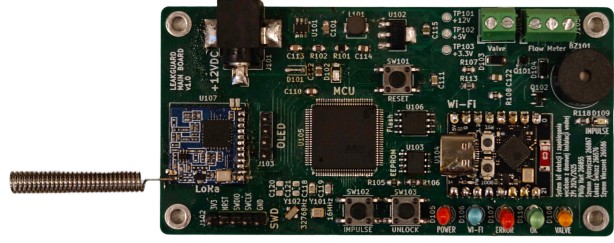

Rysunek 4.2: Zlutowany układ modułu centralnego.

zlutowano i zmontowano układy w technologii montażu powierzchniowego (SMT).

## 4.1 Moduł centralny

Moduł centralny jest elementem systemu LeakGuard odpowiedzialnym za monitorowanie przepływu wody w instalacji wodnej oraz jego blokowanie, jeżeli został wykryty wyciek. Urządzenie jest przeznaczone do zamontowania w instalacji wodnej wraz z elementami hydraulicznymi - przepływomierzem oraz elektrozaworem. Jest stale podłączone do sieci WiFi w celu umożliwienia jego konfiguracji, a także monitorowania przepływu wody w aplikacji mobilnej.

### 4.1.1 Szczegóły sprzętowe

Na rysunku 4.1 przedstawiono opracowaną płytkę drukowaną bez komponentów, a na rysunku 4.2 gotowy, zlutowany układ modułu.

Urządzenie wykorzystuje mikrokontroler STM32F746VET6 [4] [5]. Moduł ESP32 C3 Super Mini zawarty w urządzeniu jest odpowiedzialny za komunikację w sieci WiFi, sterowany poprzez protokół AT [1].

Moduł centralny został wyposażony w graficzny wyświetlacz OLED, podłączony przez interfejs I²C [10], który nie jest widoczny na rysunku. Wyświetlacz jest odpowiedzialny za przedstawianie danych kluczowych do przygotowania jednostki do pracy w sieci lokalnej, aktualny czas oraz pomiar przepływu wody.

Urządzenie zadziała z dowolnym typem elektrozaworu zasilanym napięciem 12V prądu stałego, zarówno NO[11], jak i NC[12]. Konfigurowalna przez użytkownika jest również liczba impulsów wysyłanych przez miernik przepływu wody, w przeliczeniu na jeden litr. Moduł współpracuje z każdym przepływomierzem o wyjściu sygnału typu otwarty kolektor. W trakcie prac prototypowych wykorzystano przepływomierz PM-3/4 firmy Termipol [8] oraz elektrozawór typu NO nieznanego chińskiego producenta.

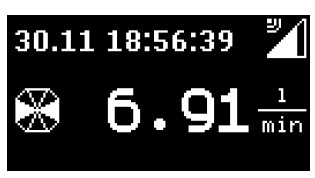

Rysunek 4.3: Przykład grafiki na wyświetlaczu OLED, która ilustruje aktualny przepływ.

Komunikacja między modułami wykorzystuje protokół LoRa. Pozwala on na bezpołączeniową komunikację bez pośrednika na relatywnie dalekie odległości. Pracuje też na niskiej częstotliwości, w paśmie 433 MHz ISM, co skutkuje lepszą zdolnością penetracji w zurbanizowanym środowisku. W fazie projektowania odrzucono takie technologie, jak ZigBee, WiFi, czy Bluetooth Low Energy, głównie ze względu na problemy z zasięgiem.

Moduł centralny posiada 64 KB pamięci EEPROM oraz 4 MB pamięci flash. W związku z większą trwałością pamięci EEPROM, znajdują się w niej często zmieniające się dane, takie jak aktualna konfiguracja systemu oraz historia przepływu wody z obecnego dnia. Natomiast w pamięci flash przechowywane są starsze dane historyczne, przedstawiane na wykresach w aplikacji mobilnej. Pozwoli to na bezawaryjną pracę urządzenia przez wiele lat.

### 4.1.2 Oprogramowanie

Oprogramowanie modułu centralnego jest wielowątkową aplikacją opartą o system operacyjny czasu rzeczywistego FreeRTOS, napisaną w języku C++. Obsługa peryferiów mikrokontrolera została zenkapsulowana w dostępnych globalnie sterownikach i usługach. Dostęp do każdej z nich wymaga pozyskania

---

[10] I²C – Inter-Integrated Circuit
[11] NO – (ang. *normally open*) normalnie otwarty
[12] NC – (ang. *normally closed*) normalnie zamknięty

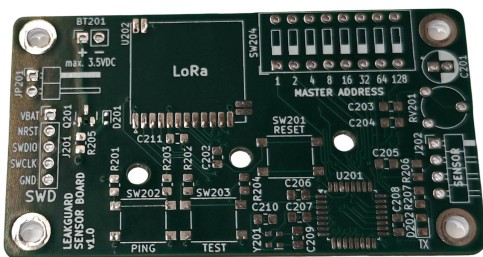

Rysunek 4.4: Gotowa płytka drukowana modułu monitora wycieków.

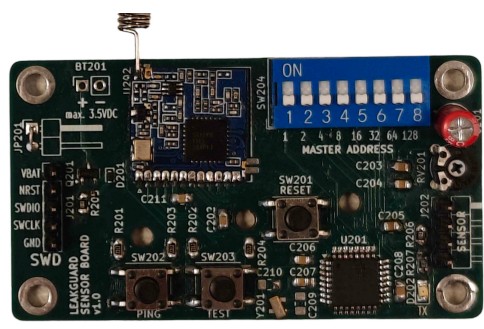

Rysunek 4.5: Zlutowany układ modułu monitora wycieków.

odpowiadającego jej muteksa, co pozwala na bezpieczną komunikację w wielozadaniowym środowisku. Priorytety zadań oraz wywłaszczanie zapobiegają wystąpieniu problemu zagłodzenia najważniejszych usług systemu.

Zaimplementowane funkcje w kodzie urządzenia obejmują:

· obsługę żądań HTTP od wielu klientów,
· periodyczne aktualizacje zegara czasu rzeczywistego przez usługę SNTP[13],
· prezentację stanu modułu na monochromatycznym wyświetlaczu OLED oraz diodach statusu,
· odbieranie i przetwarzanie danych od modułu LoRa,
· zapisywanie konfiguracji oraz danych historycznych w pamięci nieulotnej,
· zliczanie impulsów z przepływomierza wraz z konwersją na litry na minutę,
· agregację i analizę danych w celu heurystycznego określenia wystąpienia wycieku,
· blokadę zaworu na podstawie harmonogramu określonego w aplikacji mobilnej,
· sterowanie elektrozaworem.

## 4.2  Moduł monitora wycieków

Moduł monitora wycieków jest odpowiedzialny za wykrywanie obecności wody w trudno dostępnych miejscach, takich jak część instalacji wodnej podatna na uszkodzenia lub przestrzeń pod wanną. Po wykryciu zalania moduł powiadamia jednostkę centralną drogą radiową o wystąpieniu wycieku. Urządzenie jest zasilane za pomocą dwóch baterii AAA.

### 4.2.1  Szczegóły sprzętowe

Urządzenie zostało zaprojektowane w oparciu o mikrokontroler STM32L031K6T6 [6] [7], charakteryzujący się bardzo niskim zużyciem energii oraz możliwością wejścia w głębokie uśpienie (Stop Mode). Za komunikację z modułem centralnym odpowiedzialny jest układ SX1278, który korzysta z protokołu LoRa. Monitor posiada dwa rezystancyjne czujniki zalania oraz przełącznik typu DIP Switch, a także dwa przyciski - TEST oraz PING.

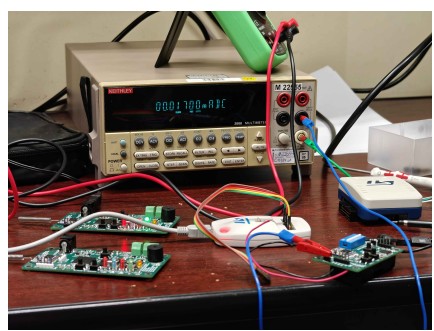

Rysunek 4.6: Pomiar poboru prądu modułu monitora wycieków za pomocą multimetru Keithley 2000.

W trakcie testów udało się osiągnąć pobór prądu w wysokości 17 µA (rys. 4.6), gdy monitor był w trybie uśpienia. Układ LoRa SX1278 pobiera około 120 mA w czasie transmisji przy maksymalnej mocy nadajnika [3]. Przy założeniu, że nadawanie odbywa się co 12 godzin, trwa pół sekundy, pojemność ogniwa AAA to 1200 mAh, a sumaryczny pobór prądu przez układ w trakcie transmisji to 150 mA, uzyskano czas życia na baterii przekraczający 7 lat, jeśli nie zostanie wykryty żaden wyciek.

$$\frac{1200\,\mathrm{mAh}}{\frac{0{,}5}{3600\cdot 12}\cdot 150\,\mathrm{mA} + (1 - \frac{0{,}5}{3600\cdot 12})\cdot 0{,}017\,\mathrm{mA}} \approx 64\,048\,\mathrm{h} \approx 7{,}3\,\mathrm{lat} \tag{4.1}$$

---

[13]SNTP – Simple Network Time Protocol

### 4.2.2 Oprogramowanie

Oprogramowanie wbudowane modułu monitora wycieków zostało opracowane w języku C. Został on wybrany ze względu na ograniczone zasoby sprzętowe oraz energooszczędność.

Moduł początkowo znajduje się w stanie uśpienia. W momencie, gdy użytkownik chce sparować monitor wycieków z modułem centralnym, naciska przycisk PING. Wtedy monitor wybudza się i kilkukrotnie nadaje pakiet przez układ SX1278. Pakiet ten składa się z typu, 96-bitowego unikalnego ID mikrokontrolera (nadanego przez producenta), 8-bitowego adresu monitora (wybranego na DIP Switchu) oraz stanu baterii. Do pakietu będzie dołączona suma kontrolna CRC, która pozwoli na weryfikację integralności. Następnie monitor przechodzi ponownie w tryb uśpienia. W przypadku, gdy przynajmniej jeden z czujników rezystancyjnych wykryje obecność wody, monitor zostanie wybudzony oraz kilkukrotnie nada do modułu centralnego pakiet informujący o wycieku. Operacja kończy się przejściem w tryb uśpienia. Ponadto, co 12 godzin monitor wybudza się z trybu uśpienia i kilkukrotnie przesyła pakiet o statusie baterii.

Wyszczególniono następujące typy pakietów:
- 00 – ping
- 01 – status baterii
- 02 – informacja o wycieku
- 03–FF – zarezerwowane

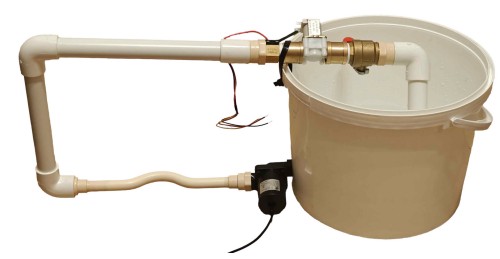

## 4.3  Testowy układ wodny

W celu testowania systemu skonstruowano instalację z zamkniętym obiegiem wody (rys. 4.7). Pozwala ona zasymulować zużycie wody w rzeczywistej instalacji domowej.

Rysunek 4.7: Testowy układ wodny. Górna część układu jest elementem systemu i służy do pomiaru oraz kontroli przepływu wody.

Układ składa się ze zbiornika, pompy odśrodkowej, przepływomierza, elektrozaworu oraz zaworu ręcznego. Pompa wytwarza maksymalny przepływ ok. 6 l/min po jej podłączeniu do typowego zasilacza do laptopów (19 V), co odpowiada średniemu przepływowi z typowego kranu. Symulacja zmiany prędkości przepływu wody w układzie testującym jest realizowana poprzez ręczne przymykanie zaworu.

Testy pokazały, że sprzęt opracowany w ramach projektu jest w stanie poprawnie monitorować zużycie wody w instalacji, wykrywać wycieki wody oraz im zapobiegać. Zaobserwowano również, że system jest w stanie mierzyć znacznie niższe przepływy wody, jak podano w nocie katalogowej przepływomierza [8]. Minimalny zmierzony przepływ wynosił około 0,1 l/min. Okazuje się to przydatne w przypadku detekcji małych, niepożądanych przepływów wody. Są to sytuacje, gdy kran nie jest całkowicie zamknięty lub wystąpiło niewielkie pęknięcie rur.

## 4.4  Aplikacja mobilna

Aplikacja mobilna LeakGuard służy jako główny interfejs użytkownika do zarządzania i monitorowania systemu. Została zaprojektowana zgodnie ze stylem neumorfizmu, co zapewnia nowoczesny i przyjazny dla użytkownika interfejs, jednocześnie zachowując wysoką funkcjonalność i czytelność.

### 4.4.1  Architektura systemu

Aplikacja wykorzystuje hierarchiczną strukturę organizacji elementów systemu:
- grupy (np. kuchnia, łazienka) - logiczne zgrupowanie jednostek centralnych,
- jednostki centralne - urządzenia monitorujące przepływ wody,
- monitory wycieku - czujniki wykrywające obecność wody w strategicznych punktach.

### 4.4.2  Komunikacja z urządzeniami

Aplikacja wykorzystuje dwa główne protokoły do komunikacji z jednostkami centralnymi:
1. mDNS (Multicast DNS) - służy do automatycznego wykrywania dostępnych jednostek centralnych w sieci lokalnej,
2. HTTP - zapewnia stabilną komunikację z jednostkami centralnymi poprzez REST API.

### 4.4.3  Główne funkcje

Aplikacja oferuje szereg funkcji zarządzania systemem:

- monitorowanie przepływu wody w czasie rzeczywistym (rys. 4.8),
- przeglądanie historii zużycia wody w formie wykresów kolumnowych (rys. 4.8),
- konfiguracja parametrów detekcji wycieków,
- zarządzanie grupami i jednostkami centralnymi (rys. 4.9, 4.10 i 4.11)
- powiadomienia o wykrytych wyciekach,
- zdalne blokowanie przepływu wody (rys. 4.8),
- konfiguracja harmonogramów blokowania przepływu (rys. 4.8).

### 4.4.4 Bezpieczeństwo

Wymiana danych między aplikacją a jednostkami centralnymi odbywa się wyłącznie w sieci lokalnej, co zapewnia podstawowy poziom bezpieczeństwa. W celu umożliwienia użytkownikowi otrzymywania powiadomień o wyciekach poza siecią domową, system wykorzystuje usługi chmurowe do przesyłania powiadomień push. Ta funkcja została zaimplementowana z wykorzystaniem platformy Firebase Cloud Messaging, zapewniając szybkie dostarczanie alertów o wykrytych wyciekach niezależnie od lokalizacji użytkownika. Powiadomienia FCM są generowane przez infrastrukturę wdrożoną na chmurze AWS, która obsługuje również brokera MQTT.

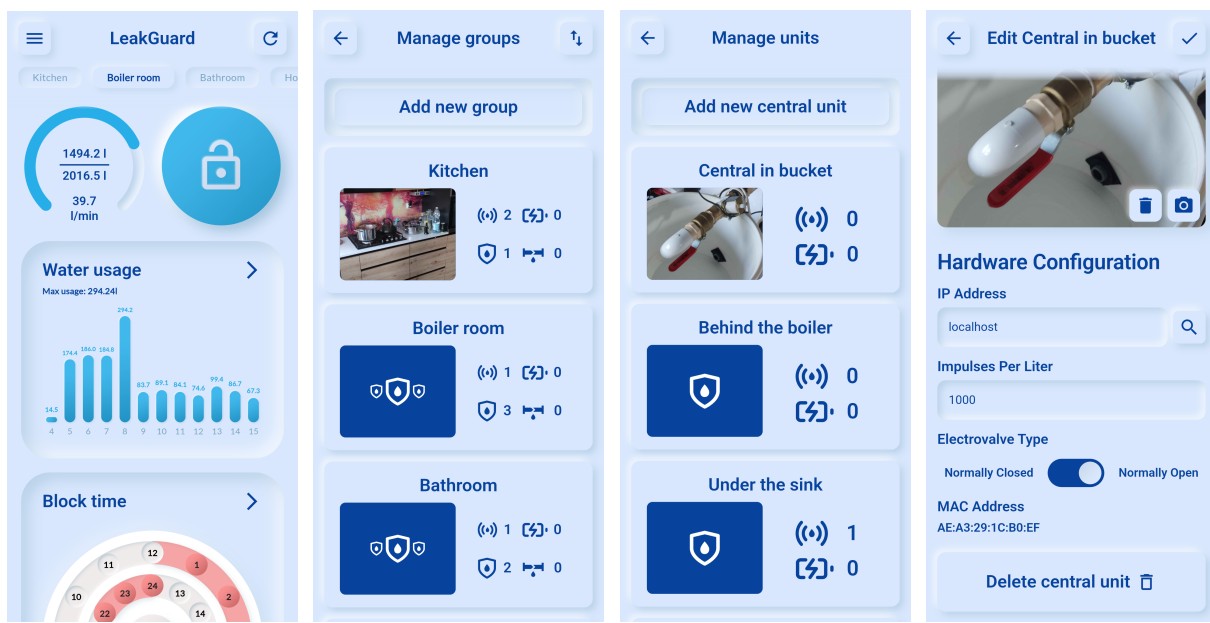

Rysunek 4.8: Ekran główny aplikacji

Rysunek 4.9: Ekran zarządzania grupami

Rysunek 4.10: Lista jednostek centralnych

Rysunek 4.11: Ekran edycji jednostki centralnej

## 5 PODSUMOWANIE I WNIOSKI

W ograniczonym do 10 tygodni czasie projektu udało się spełnić wszystkie wymagania funkcjonalne i niefunkcjonalne. Stworzono rozwiązanie, które jest w stanie efektywnie zapobiegać skutkom wycieków z instalacji wody użytkowej.

### 5.1 Osiągnięcia

Jednym z największych osiągnięć projektu jest optymalizacja zużycia energii przez monitor wycieku. Dzięki wybraniu odpowiedniego mikrokontrolera oraz zoptymalizowaniu kodu użytkownik systemu będzie zmuszony do wymiany baterii co 7 lat.

Zbudowanie testowego systemu wodnego pozwoliło na przetestowanie całego produktu w warunkach zbliżonych do rzeczywistej instalacji wody użytkowej. Umożliwiło to upewnienie się, że warunki środowiskowe, takie jak interferencja elektromagnetyczna, nie będą zakłócać działania systemu.

Kod systemu napisany został na tyle modularnie, że każdy z członków zespołu mógł pracować nad swoją częścią niezależnie. Dzięki zastosowaniu narzędzia CMake, integracja modułów przebiegała całkowicie bezproblemowo.

## 5.2 Perspektywy dalszego rozwoju projektu

W projekcie istnieje możliwość wprowadzenia kilku dalszych udoskonaleń, które by pozwoliły wdrożyć produkt na rynek:

**Szczelne obudowy do modułów monitorów wycieków** Wydrukowane w 3D i uszczelnione obudowy zapewnią ochronę elektroniki przed uszkodzeniem spowodowanym wodą.

**Zwiększone bezpieczeństwo transmisji radiowych** Można zaimplementować szyfrowanie i numerowanie wiadomości wysyłanych modułami LoRa, wykorzystując sprzętowe peryferia szyfrujące, tak by zapobiec atakom np. typu replay.

**Usprawnienie bezpieczeństwa dostępu** Proste uwierzytelnianie za pomocą hasła pozwoli zabezpieczyć moduły centralne przed użytkownikami, którzy nie powinni mieć dostępu do systemu.

**Zdalna kontrola zaworem spoza sieci lokalnej** Funkcja ta umożliwi sterowanie zaworami podczas nieobecności właściciela domu, co może okazać się przydatne w trudnych sytuacjach.

## 5.3 Podziękowania

Składamy serdeczne podziękowania mgr. inż. Piotrowi Jóźwiakowi, promotorowi zespołu LeakGuard, za oferowane wsparcie oraz pozytywną atmosferę podczas rozwijania projektu. Wyrażamy również głęboką wdzięczność Panu Rafałowi Wieczorkowi, który oferował nieocenioną pomoc podczas opracowywania i konstrukcji testowego układu wodnego. Nie moglibyśmy ukończyć tej pracy bez hojnej pomocy firm b:art instruments i Dolvac Instruments, a także Wydziału Elektroniki, Fotoniki i Mikrosystemów Politechniki Wrocławskiej, dzięki którym mogliśmy korzystać ze sprzętu potrzebnego do skonstruowania modułów oraz dokonania kluczowych pomiarów. Wyrażamy również uznanie dla obserwatorów nieformalnych demonstracji projektu we wczesnych etapach rozwoju, którzy oferowali wnikliwe spojrzenie na pożądane cechy ostatecznego produktu. Jesteśmy wdzięczni wszystkim, którzy poświęcili swój czas, wiedzę i wsparcie w trakcie realizacji projektu LeakGuard.

## LITERATURA

[1] Espressif. *AT Command Set.* `https://docs.espressif.com/projects/esp-at/en/release-v3.3.0.0/esp32/AT_Command_Set/index.html` [dostęp 30.11.2024].

[2] Maciej Knapik. Analysis of the cost of leaks from the water supply system, including methods to reduce water consumption. *Zeszyty Naukowe SGSP/Szkoła Główna Służby Pożarniczej*, (84), 2022.

[3] Semtech Corporation. *SX1276/77/78/79 - 137 MHz to 1020 MHz Low-Power Long Range Transceiver*, maj 2020. Rev 5 `https://semtech.my.salesforce.com/sfc/p/#E0000000JelG/a/2R0000001Rc1/QnUuV9TviODKUgt_rpBlPz.EZA_PNK7Rpi8HA5..Sbo` [dostęp 29.11.2024].

[4] STMicroelectronics. *STM32F745xx, STM32F746xx datasheet*, luty 2016. DocID027590 Rev 4 `https://www.st.com/resource/en/datasheet/stm32f746ve.pdf` [dostęp 29.11.2024].

[5] STMicroelectronics. *STM32F75xxx and STM32F74xxx advanced Arm®-based 32-bit MCUs*, czerwiec 2018. RM0385 Rev 8 `https://www.st.com/resource/en/reference_manual/rm0385-stm32f75xxx-and-stm32f74xxx-advanced-armbased-32bit-mcus-stmicroelectronics.pdf` [dostęp 29.11.2024].

[6] STMicroelectronics. *STM32L031x4, STM32L031x6 datasheet*, marzec 2018. DS10668 Rev 6 `https://www.st.com/resource/en/datasheet/stm32l031k6.pdf` [dostęp 30.11.2024].

[7] STMicroelectronics. *Ultra-low-power STM32L0x1 advanced Arm®-based 32-bit MCUs*, luty 2022. RM0377 Rev 8 `https://www.st.com/resource/en/reference_manual/rm0377-ultralowpower-stm32l0x1-advanced-armbased-32bit-mcus-stmicroelectronics.pdf` [dostęp 30.11.2024].

[8] Termipol. *Przepływomierz PM3/4-B*, 2023. `https://termipol.pl/media/products/3b828f9209a368fcbf01093ecdc06723/attachments/pl_PL/karta-katalogowa-przeplywomierz-pm3-4-b.pdf` [dostęp 29.11.2024].
