# OpenReview forum: "LeakGuard - System IoT detekcji i zapobiegania wyciekom z domowej instalacji wodnej"
_pwr.edu.pl/Wrocław_University_of_Science_and_Technology/2024/ZPI_Day — Wrocław University of Science and Technology 2024 ZPI Day Submission_

### Official Review · Reviewer_fECR · 2024-12-04
**Dojrzały projekt inżynierski wykraczający poza dziedzinę informatyki**

**Confidence:** 5
**Significance Of Results:** 5
**Overall Quality:** 4

**Compliance With Template:**

5: Very High Quality – The article contains all the required sections, which are written in a very detailed, clear, and error-free manner. The structure is professional and meets expectations, and the content adheres to the highest substantive and formal standards.

**Description Of Results:**

5: Very High Quality – The results are described in detail, clearly and comprehensively, supported by thorough evaluation, analysis, and convincing usage examples. The description meets the highest substantive standards.

**Feedback On Consistency:**

+ Artykuł jest napisany zgodnie ze sztuką. Można odnaleźć niewielkie błędy stylistyczne, wynikające z małego doświadczenia Autorów w pisaniu tego typu opracowań. Treść jest czytelna, dobrze podzielona na rozdziały. Dostarczony opis jest wystarczająco szczegółowy, aby czytelnik potrafił wyobrazić sobie system jako całość z zachowaniem limitu wielkości dokumentu. Dokument nie posiada zbędnych powtórzeń tekstu pomiędzy rozdziałami.

- Brakuje opisu algorytmu wykrywania wycieku na jednostce centralnej.

**Potential For Development:**

+ Projekt systemu wykazuje dużą dojrzałość, pozwalającą w niedługim czasie komercjalizować rozwiązanie.

+ Zaproponowane usprawnienia są celne i warte dalszego rozwijania.

**Project Nature Evaluation:**

+ Opisany projekt znacząco wybiega poza wymogi pracy inżynierskiej z dziedziny informatyki. Autorzy zademonstrowali szeroką wiedzę zarówno z informatyki jak i elektroniki. Przekrój wykorzystanych technologii oraz ich dobór jakościowy potwierdza wysoką dojrzałość Autorów jako sprawnych inżynierów. W ramach projektu opracowano dwa urządzenia elektroniczne wraz z oprogramowaniem, opracowano aplikację mobilną do zarządzania systemem oraz wykonano integrację w usługami chmurowymi AWS. Zastosowano nowoczesne rozwiązania, jak choćby technologię LoRA do komunikacji pomiędzy urządzeniem centralnym i monitorami wycieku. Całość bardzo dobrze wpisuje się w świat Internetu Rzeczy (IoT) oraz technologii inteligentnych budynków.

+ Opracowany moduł monitora wycieku pracujący na baterii, według autorów nawet do 7 lat, jest znaczącym osiągnięciem.

**Technical Language Precision:**

4: High Quality – The language is appropriate for a technical report. Terminology is used correctly, and statements are precise, with only minor shortcomings that do not affect the overall clarity.

---

### Official Review · Reviewer_FAXq · 2024-12-06
**Recenzja dla projektu LeakGuard**

**Confidence:** 4
**Significance Of Results:** 5
**Overall Quality:** 5

**Compliance With Template:**

5: Very High Quality – The article contains all the required sections, which are written in a very detailed, clear, and error-free manner. The structure is professional and meets expectations, and the content adheres to the highest substantive and formal standards.

**Description Of Results:**

4: High Quality – The results are described in detail and supported by usage examples or evaluations. The description is reliable but may lack full depth of analysis.

**Feedback On Consistency:**

Artykuł napisany jest w sposób przejrzysty i zrozumiały. Co istotne, wolny jest w zasadzie od kolokwializmów oraz anglicyzmów. Ze względu na bardzo ograniczoną obiętość zrozumiałym jest, że trudno umieścić w artykule definicje wszystkich użytych terminów oraz opisy wykorzystywanych urządzeń i elementów, natomiast na pewno można byłoby przyjąć jednolity dla całej pracy sposób zapisu nazw, np. Internet Rzeczy (ang. Internet of Things, IOT). W kilku miejscach zapisy różnią się między sobą (np. wspomniane IoT oraz wyjście przepływomierza typu otwarty kolektor). Uwaga ta ma raczej na celu zwrócenia uwagi autorów na ten aspekt przy przygotowywaniu kolejnych publikacji i nie wpływa na pozytywny odbiór pracy.
Struktura pracy jest spójna i zgodna z wymaganiami. Jedyna uwaga dotyczy możliwości szerszego przedstawienia wyników testów przygotowanego rozwiązania, które można byłoby bardziej wyróżnić w tekście pracy oraz zamieścić dyskusję przypadków brzegowych - np. przeprowadzenia symulacji zużycia prądu dla wspomnianych w tekście niewielkich wycieków (odpowiadając tym samym na pytanie, czy urządzenie nie będzie wzbudzane zbyt często).

**Potential For Development:**

W artykule zostały przedstawione 4 możliwości dalszego rozwoju przedstawionego rozwiązania. Dotyczą one aspektów bezpieczeństwa i niezawodności (odporność na zalania, szyfrowanie komunikacji, uwierzytelnienie) oraz dostępności poza siecią lokalną. Szczególnie ostatni element uważam za wartościowy ze względu na naturę projektu, ponieważ możliwość zdalnego monitorowania instalacji wodnej i powiadamiania użytkownika na odległość o wyciekach wydaje się być bardziej istotna niż samo ograniczanie zużycia wody. Autorzy powołują się na publikację M.Knapika, według której koszty pochodzące z niezidentyfikowanych wycieków dochodzą do 1700zł. Kwota ta, choć wysoka, może być znacząco niższa niż straty spowodowane zalewaniem mieszkań podczas wielodniowych nieobecności właścicieli.
W kwestii potencjalnego rozwoju warto byłoby rozważyć konieczne usprawnienia, które należałoby wprowadzić przy adaptacji rozwiązania do środowisk innych niż domowe (o czym Autorzy wspominają w pierwszej części pracy). Dodatkowo można byłoby pomyśleć o dodaniu dodatkowych modułów, jak np. mierniki ciśnienia czy jakości wody.

**Project Nature Evaluation:**

Projekt ma jasno zdefiniowany cel, a wybrane narzędzia i rozwiązania technologiczne są adekwatne w omawianej dziedzinie. Doceniam przede wszystkim umieszczenie w tekście pracy wszystkich użytych modułów, elementów elektronicznych oraz protokołów. W mojej ocenie projekt mógłby z powodzeniem znaleźć zastosowanie praktyczne, zgodne z załóżonym celem.
Na pewno pozytywny wpływ na odbiór pracy miałoby umieszczenie ogólnej architektury opracowanego rozwiązania, z zaznaczeniem wszystkich połączeń i użytych protokołów. Ponownie, zdaje sobie sprawę, że ze względu na organiczoną objętność artykułu należało wybrać elementy najbardziej istotne (a bez wątpienia są nimi opisy wykonanych układów oraz oprogramowania), niemniej przy przygotowywaniu dokumentacji projektowej warto byłoby rozważyć ten element.

**Technical Language Precision:**

4: High Quality – The language is appropriate for a technical report. Terminology is used correctly, and statements are precise, with only minor shortcomings that do not affect the overall clarity.

---

### Official Review · Reviewer_RbEd · 2024-12-09
**LeakGuard - System IoT detekcji i zapobiegania wyciekom z domowej instalacji wodnej**

**Confidence:** 4
**Significance Of Results:** 3
**Overall Quality:** 3

**Compliance With Template:**

3: Average Quality – The article includes most of the required sections, but some may be incomplete, written in a general or unclear manner. The content is correct but requires further refinement.

**Description Of Results:**

3: Average Quality – The results are described with moderate detail. Some examples or evaluation elements are present but insufficiently developed or incomplete.

**Feedback On Consistency:**

Opis projektu jest spójny, a prezentowane informacje są w większości logicznie powiązane.
Brak jest informacji w jakim odniesieniu/skali zostały przytoczone badania rocznych kosztów niezidentyfikowanych wycieków na poziomie wartości 1700zł.
Szacowanie obliczeniowe podtrzymania monitora wycieku na ogniwie/baterii niesprecyzowanej technologii  jest daleko optymistyczne przy pominięciu aspektu samorozładowania tegoż źródła energii i tolerancji/błędów przyjętych wartości obliczeniowych oraz warunków środowiskowych. Wobec błędnego oszacowania może zostać zawiedzione zaufanie przyszłego ewentualnego klienta do prezentowanych parametrów.
Woda w rzeczywistej instalacji domowej oprócz zakresu przepływu jest pod ustandaryzowanym przez dostawcę ciśnieniem (np. 0.5-4.5 bar), pomiary w środowisku testowym z kolei opatrzone są jedynie informacją o maksymalnej wydajności pompy, co bez ustalenia ciśnienia przepływu może się okazać nie bez znaczenia dla prezentowanych wyników.
Brak jest w pracy informacji czy wobec znalezienia kluczowego błędu w specyfikacji technicznej producenta przepływomierza zostały podjęte próby wyjaśnienia tego stanu z przedstawicielem producenta?
Jako osiągnięcie podano:...upewnienie się, że warunki środowiskowe, takie jak interferencja elektromagnetyczna nie będą zakłócać działania systemu...", brakuje tutaj wskazania w jakim ujęciu zostało to przeprowadzone, jakim sposobem ustalone.

**Potential For Development:**

Artykuł wskazuje jakie aspekty projektu wymagają rozwoju w przyszłości. Poprawienie metod pomiarowych i warunków testowych może przynieść potencjał zastosowań komercyjnych. Niestety kluczowe dla wyników wyzwania i bariery zostały pominięte co zostało pokrótce wymienione powyżej.

**Project Nature Evaluation:**

Projekt ma wyraźny cel inżynierski, zastosowane metody oraz rozwiązania są odpowiednie do realizacji tego celu, ale wskazane powyżej aspekty wymagają poprawy, wobec nieścisłości może się okazać, że rzeczywiste wyniki są mniej optymistyczne.

**Technical Language Precision:**

4: High Quality – The language is appropriate for a technical report. Terminology is used correctly, and statements are precise, with only minor shortcomings that do not affect the overall clarity.

---

### Decision · Program_Chairs · 2024-12-10

Accept (Poster)